# An in silico testbed for fast and accurate MR labeling of orthopedic implants

Gregory M Noetscher[1]*[†], Peter J Serano[2†], Marc Horner[2], Alexander Prokop[3], Jonathan Hanson[4], Kyoko Fujimoto[5], James Brown[6], Ara Nazarian[7], Jerome Ackerman[8,9], Sergey N Makaroff[1,9]

[1]Electrical & Computer Eng. Dept, Worcester Polytechnic Institute, Worcester, United States; [2]Ansys, Canonsburg, United States; [3]Dassault Systèmes Deutschland GmbH, Darmstadt, Germany; [4]Neva Electromagnetics, LLC, Holden, United States; [5]GE HealthCare, Chicago, United States; [6]Micro Systems Enigineering, Inc, an affiliate of Biotronik, Lake Oswego, United States; [7]Musculoskeletal Translational Innovation Initiative, Department of Orthopedic Surgery, Beth Israel Deaconess Medical Center and Harvard Medical School, Boston, United States; [8]Harvard Medical School, Boston, United States; [9]Athinoula A Martinos Center for Biomed. Imaging, Massachusetts General Hospital, Charlestown, United States

*For correspondence:
gregn@wpi.edu

[†]These authors contributed equally to this work

**Abstract** One limitation on the ability to monitor health in older adults using magnetic resonance (MR) imaging is the presence of implants, where the prevalence of implantable devices (orthopedic, cardiac, neuromodulation) increases in the population, as does the pervasiveness of conditions requiring MRI studies for diagnosis (musculoskeletal diseases, infections, or cancer). The present study describes a novel multiphysics implant modeling testbed using the following approaches with two examples: (1) an in silico human model based on the widely available Visible Human Project (VHP) cryo-section dataset; (2) a finite element method (FEM) modeling software workbench from Ansys (Electronics Desktop/Mechanical) to model MR radio frequency (RF) coils and the temperature rise modeling in heterogeneous media. The in silico VHP-Female model (250 parts with an additional 40 components specifically characterizing embedded implants and resultant surrounding tissues) corresponds to a 60-year-old female with a body mass index of 36. The testbed includes the FEM-compatible in silico human model, an implant embedding procedure, a generic parameterizable MRI RF birdcage two-port coil model, a workflow for computing heat sources on the implant surface and in adjacent tissues, and a thermal FEM solver directly linked to the MR coil simulator to determine implant heating based on an MR imaging study protocol. The primary target is MR labeling of large orthopedic implants. The testbed has very recently been approved by the US Food and Drug Administration (FDA) as a medical device development tool for 1.5 T orthopedic implant examinations.

## eLife assessment

This manuscript will provide a **valuable** method to evaluate the safety of MR in patients with orthopaedic implants, which is required in clinics. A strength of the work is that the in-silicon testbed is **solid**, based on the widely available human project, and validated. In addition, the toolbox will be open for clinical practice.

## Introduction

One limitation on the ability to monitor health in older adults using magnetic resonance (MR) imaging studies is the presence of implants, where the prevalence of implantable devices (orthopedic, cardiac, neuromodulation) increases in the population, as does the pervasiveness of conditions requiring MRI

studies for diagnosis (musculoskeletal conditions, infections, or cancer). In 2020, 26% of the US population over 65 was estimated to carry a large joint or spinal implant (*Kanal et al., 2015*). Simultaneously, 12.6 million patients over 65 who carry orthopedic or cardiac implants will need an MR study within 10 years, according to an estimate in 2020 (*Kanal et al., 2015*) with this number expected to rise. Similarly, over 70% of the estimated 3 million pacemakers in the US are implanted in patients older than 65 (*Lim et al., 2019*; *Puette et al., 2022*), where approximately 20% of these patients will need an MR study within 12 months of device implantation (*Brown et al., 2019*).

Terms to be used to label MR information for medical devices – implants – include *MR safe*, *MR conditional*, and *MR unsafe* (*US Department of Health and Human Services et al., 2021*; *US Food and Drug Administration, 2023a*; *American College of Radiology, 2020*; *Shellock et al., 2009*). MR safe items are nonconducting, nonmetallic, and nonmagnetic items, such as a plastic Petri dish (*Shellock et al., 2009*). MR unsafe items include in particular ferromagnetic materials (*Shellock et al., 2009*); they should not enter the MR scanner room (*US Food and Drug Administration, 2023a*). All other devices that contain any metallic components, such as titanium (regardless of ferromagnetism), are *MR conditional* and will need to be evaluated and labeled for radio frequency (RF)-induced heating, image artifact, force, and torque (*US Department of Health and Human Services et al., 2021*). For the corresponding labeling icons, see *US Food and Drug Administration, 2023a*.

MR conditional implants may safely enter the MR scanner room only under the very specific conditions provided in the labeling. Patients should not be scanned unless the device can be positively identified as MR conditional and the conditions for safe use are met (*US Food and Drug Administration, 2023a*). When present, information about expected temperature rise and artifact extent may inform the risk/benefit decision of whether a patient should or should not undergo an MR examination (*US Food and Drug Administration, 2023a*).

Given the large numbers of implants subject to conditional labeling, the number of cleared US Food and Drug Administration (FDA) 510(k) submissions for orthopedic implantable devices with MR labeling has been growing exponentially since 2014 (*Fujimoto et al., 2020*), approaching 100 in 2019 (*Fujimoto et al., 2020*). However, practical testing is limited by constraints related to cost and resources, including testing tools (*Fujimoto et al., 2020*). As a result, a number of implants have been labeled '*MR Not Evaluated*', which precludes patients' access to MR imaging procedures (*Fujimoto et al., 2020*). Other implants may be labeled too restrictively (*Kanal et al., 2015*), limiting patient access to MR imaging (*US Food and Drug Administration, 2022*). When estimating combined data from *Kanal et al., 2015*; *Fujimoto et al., 2020*; *US Food and Drug Administration, 2022*; *US Food and Drug Administration, 2023b*, up to 2 million elderly patients in the US are potentially affected by MR labeling uncertainty.

Presumably, the most important consequence of this uncertainty is restricting general access to MR imaging studies for patients with implants. This also prevents the use of MR imaging for better soft tissue monitoring in the vicinity of implants. A prime example of the latter is periprosthetic joint infection following total hip replacement surgery, which occurs in only 1–2% of primary arthroplasties (*Fischbacher and Borens, 2019*; *Zanetti, 2020*) but in up to 30% of revision arthroplasties (*Zanetti, 2020*). This form of infection occurs due to mechanical loosening and dislocation, currently the most common causes for revision of total hip arthroplasty in the US (*Li and Glassman, 2018*). Periprosthetic joint infection-related mortality is approaching 5–8% at 1 year (*Fischbacher and Borens, 2019*). Presently, X-rays and other methods are used for diagnosis (*Sheth et al., 2023*), but results of MR imaging with metal artifact reduction were recently shown to be the most accurate tool in the diagnosis of several biomarkers of periprosthetic hip joint infection (*Zanetti, 2020*; *Galley et al., 2020*).

One major safety concern, relevant to both passive and active implants, is implant heating within MR RF and gradient coils (*US Department of Health and Human Services et al., 2021*; *Song et al., 2018*; *Al-Dayeh et al., 2020*; *Winter et al., 2021*; *Wooldridge et al., 2021*; *Arduino et al., 2021*; *Arduino et al., 2022a*; *Bassen and Zaidi, 2022*; *Arduino et al., 2022b*; *Clementi et al., 2022*; *Oberle, 2021*). Along with the required yet not entirely anatomical ASTM phantom test and other similar phantom tests (*US Department of Health and Human Services et al., 2021*; *Song et al., 2018*; *Winter et al., 2021*; *Wooldridge et al., 2021*; *Bassen and Zaidi, 2022*; *Arduino et al., 2022b*), numerical simulations with virtual human models generate accurate predictions of temperature rise (*Al-Dayeh et al., 2020*; *Winter et al., 2021*; *Wooldridge et al., 2021*; *Arduino et al., 2022a*; *Clementi et al., 2022*; *Oberle, 2021*) accepted by the FDA (*Oberle, 2021*). The electromagnetic and

thermal simulation algorithms based on finite element, finite difference, and boundary element methods are reasonably well developed (*Makarov et al., 2017*; *Noetscher et al., 2023*; *Noetscher et al., 2021*; *Noetscher, 2021*). However, accessible, full body, detailed anatomical virtual human models reflecting major age, sex, race, and obesity variations are severely lacking. Their creation is a long, tedious, and labor-intensive process. Even today, it requires manual and semiautomatic supervised segmentation of full body MR images, surface mesh reconstruction, mesh intersection resolution, software compatibility and robustness testing, and finally examination of hundreds of different body compartments by anatomical experts.

An excellent collection of in silico human body models intended for this purpose is the Virtual Population, a product of the IT'IS Foundation (*Oberle, 2021*; *Gosselin et al., 2014*) widely used in both industrial and academic applications. While this population has many highly detailed body models, it is relatively homogeneous: reasonably fit, younger Caucasian European subjects, representing a comparatively limited subsection of human anatomy and physiology. Although three obese models were added in 2023 (*IT'IS Foundation, 2023*), two of the three are not truly anatomical and were obtained via morphing (the 'Fats' model being the exception). Also, while models are available for purchase and for research purposes, no background MRI data enabling independent tissue structure verification have been made publicly available for this population set.

The present study describes a complete ready-to-use implant modeling testbed for RF heating based on:

1. an in silico human model constructed from the widely available Visible Human Project (VHP) (*Spitzer et al., 1996*; *Ackerman, 1998*) cryo-section dataset;
2. a FEM modeling software workbench from Ansys HFSS (Electronics Desktop) to model the physical phenomena of an MR RF coil and corresponding temperature rise in heterogeneous media.

The in silico VHP-Female model (250 anatomical structures with an additional 40 components specifically modeling embedded implants *Noetscher et al., 2021*) characterizes a 60-year-old female subject with a body mass index of 36. The open-source version of this model (*NEVA Electromagnetics, LLC, 2015*) has over 600 registered users from both industry and academia worldwide. The testbed includes the in silico model, an implant embedding procedure, a generic parameterizable MR RF birdcage two-port coil model, a workflow for computing heat sources on the implant surface and neighboring tissues, and a thermal FEM solver directly linked to the MR coil simulator to estimate implant heating based on an MR imaging protocol. The primary target is MR labeling of large orthopedic implants. The testbed has recently been approved by the FDA as a medical device development tool (*Neva Electromagnetics, LLC, 2022*) for 1.5 T orthopedic implant examinations. We also present two simple application examples pertinent to choosing an appropriate MR imaging protocol for a particular orthopedic implant as well as validation against measurements of the heading of an ablation needle in bovine liver.

## Materials and methods
### In silico human model and implant embedding procedure
*Figure 1* shows surface CAD meshes for the VHP-Female model (*Makarov et al., 2017*; *Noetscher et al., 2021*) (with some muscles removed for clarity) and examples of passive femoral implants embedded into the model. The corresponding physical femoral implants are shown on the top right of the figure.

The implant registration enforces an anatomically correct implant position, and a certain part of the bone matter (cortical and/or trabecular) to be removed as necessary. A semiautomatic implant registration algorithm requiring limited user intervention has been employed based on the principal idea to use at least two anchor nodes per implant: a fixed node and a floating node. The floating anchor node is a vertex of the implant mesh belonging to a certain curve, say, the long axis of the bone. The fixed anchor node is a joint coincident vertex of the femur mesh and the implant mesh. These nodes define the proper implant position given the bone model and a cost function, with a 'best fit' based on a mesh intersection check and the signed normal distances between implant/bone boundaries. An additional criterion involves the minimum required thickness of the cortical bone matter with an embedded implant.

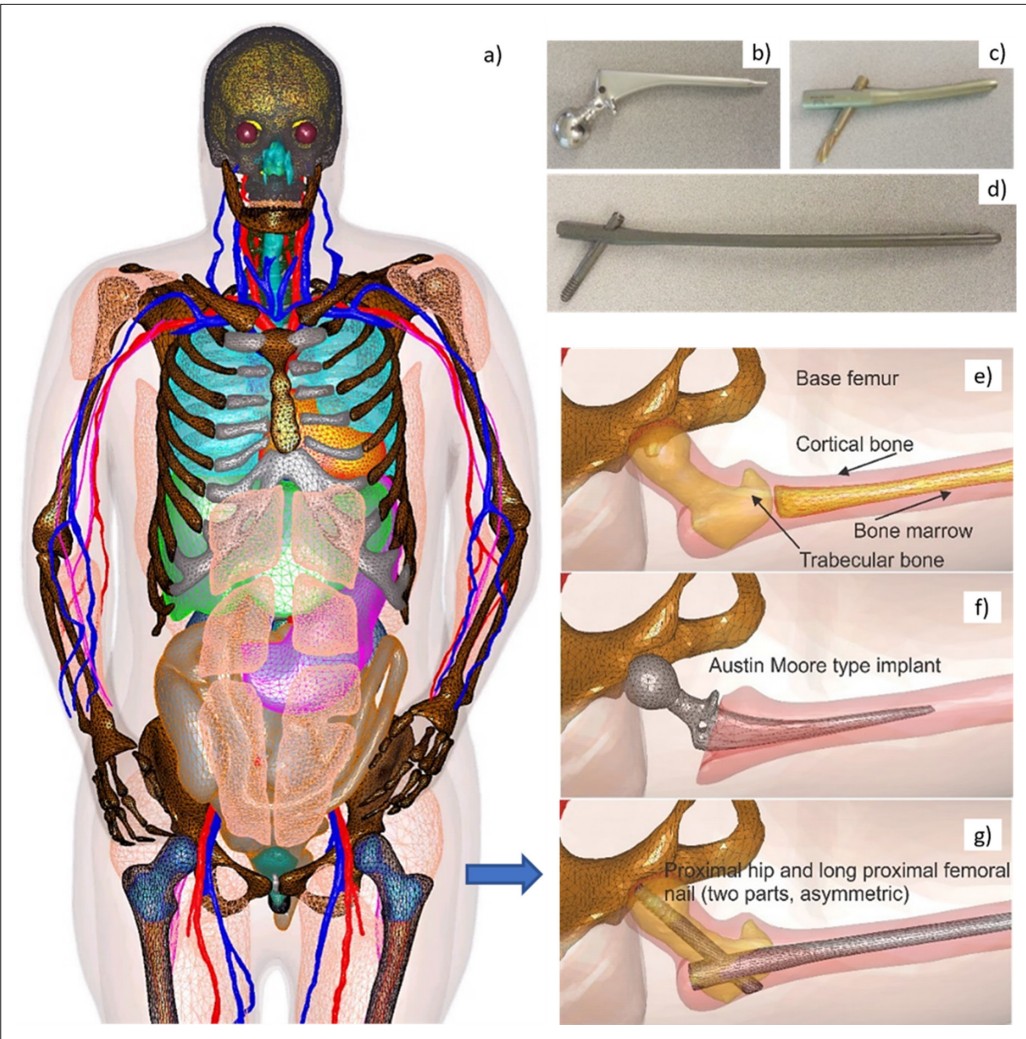

**Figure 1.** Visible Human Project (VHP)-Female Model with Embedded Passive Implants. Left (**a**) – surface CAD meshes for the Visible Human Project (VHP)-Female model (with some muscles removed for purposes of visualization); and right – examples of passive femoral implants embedded into the model. (**b–d**) At top right – physical femoral implants; (**e–g**) at center and bottom – anatomically justified CAD realization within the virtual human VHP-Female. An Austin Moore implant is shown in **b**; a short proximal femoral nail with the proximal hip (a large femoral neck screw) is given in **c**; a long proximal femoral nail with the proximal hip is presented in **d**.

## Computation of heat sources due to microwave absorption in MR RF coils

A generic, parameterized, and tunable MR RF birdcage two-port coil model (high-, low-, or bandpass), at 64 MHz (1.5 T) with a variable number of rungs was implemented in Ansys Electronics Desktop (Ansys HFSS, *Figure 2a*). This model is used to compute heat sources – either specific absorption rate (SAR) in W/kg or power loss density in W/m³ at any point in the body, including on the surface of the implants (*Figure 2b, c*; *Kozlov et al., 2015*). The in silico model with the implant(s) can be positioned at any appropriate landmark.

## Determination of implant temperature rise as a function of scan time

An Ansys FEM transient thermal solver was employed to determine tissue temperature rise close to the implant caused by the heat sources. It requires knowing the relevant thermal properties of the tissues. The solver may approximately model blood perfusion, which is less important for bone, but is important for cardiac implants and other soft tissue implants (*Winter et al., 2021*).

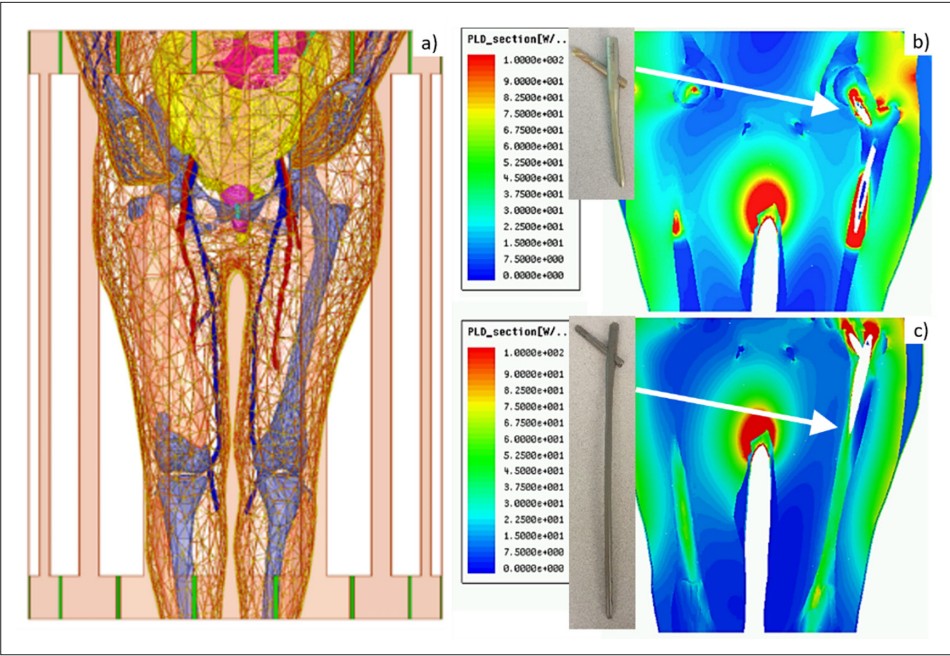

**Figure 2.** Left – Visible Human Project (VHP)-Female computational phantom positioned within a 1.5 T MRI birdcage coil at the abdominal landmark. Right – power loss density in W/m³ in the coil for the (**b**) Austin Moore and (**c**) femoral nail implants.

The entire testbed has been integrated into Ansys Workbench, which allows the combination of different multiphysics modules within a single environment, as shown in *Figure 3*.

In this way, the output of the electromagnetic solver (*Figure 3A and B*) is the input to the thermal solver (*Figure 3D*). Both accurate FEM solvers utilize the same human model geometry throughout, but with different material properties (thermal vs electromagnetic).

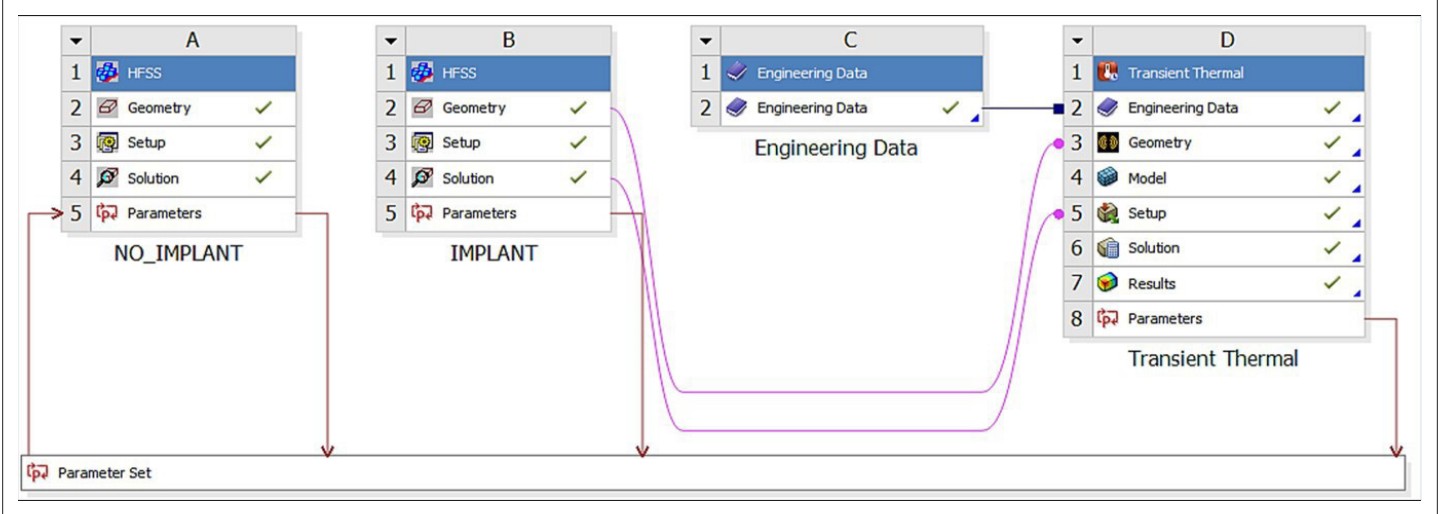

**Figure 3.** Ansys Workbench modeling workflow consisting of the HFSS (electromagnetic) module labeled as A and B, and the transient thermal module labeled D. Thermal material properties are contained in module C.

## Results

### What is heating-related MR labeling of implants?

For the implantable devices that are categorized as MR conditional, the labeling includes scan and rest times at a given whole-body SAR or $B^+_{1, rms}$ ('root mean square' value of $B^+_1$ averaged over a period of 10 s). This is described in the FDA's guidance document (*US Department of Health and Human Services et al., 2021*) (cf. also examples in *Shellock et al., 2009*) where devices are to be labeled for a 1 hr MR session, including both scan and rest times. The guidance states a certain interleaving combination of scan (e.g., 5 min) and rest (e.g., 15 min) times that guarantees implant heating is less than 5°C or another specified number (*US Department of Health and Human Services et al., 2021*). The FDA-required procedure is a measurement test in an ASTM gel-based homogeneous phantom (*US Department of Health and Human Services et al., 2021*; *Song et al., 2018*). When performing relevant numerical modeling, the pulse sequences and scan times should be converted to equivalent CW (continuous wave) operation, which is easier to model.

However, the response of the ASTM phantom is quite different from that of a real body, which includes bones and other tissues of varying electrical and thermal conductivities, as well as blood circulation and perfusion. In several test cases, our testbed prototype predicted a higher maximum temperature rise (up to 40% higher) at the implant tips versus in vitro experiments with a simplified gel phantom. In other cases, and for other implants, however, the heating was substantially lower (by 50% or so). Therefore, the in silico testbed will augment the ASTM measurements with accurate multiphysics modeling. Additionally, this modeling can assist with implant design in an efficient manner.

We note that this work is solely focused on the RF safety aspects of MR labeling. The results expressed herein should be considered supplemental to existing published guidelines.

### Example: labeling long femoral titanium nail

*Figure 4* shows an example of testing results for a long titanium femoral nail subject to three cycles of 15 min with a 2.3 W/kg average equivalent SAR exposure followed by 5 min of rest, resulting in a 1 hr total exposure in a 1.5 T MRI coil. The model predicts that the temperature near the implant reaches 41°C after the first exposure with its final value approaching 45°C, a total increase of about 10°C which is clearly unacceptable! Further simulations show that 4 min exposures followed by 16 min of rest would be a safe solution. In the testbed, the exposure time is arbitrary and can be rapidly tested and adjusted (within 5–7 min) to meet the FDA requirements (*US Department of Health and Human Services et al., 2021*) and construct the proper MR exposure protocol.

### Comparison with experiment for a long resonant metal conductor with a sharp tip

The most challenging and important cases correspond to testing large and potentially resonant metal implants (*Song et al., 2018*) with relatively sharp tips or terminations since detecting resonance requires accurate high-frequency modeling. One extreme example was studied in *Hue et al., 2018*, where a simulated percutaneous RF ablation surgical procedure using MR heating was performed in ex vivo bovine liver in a 1.5 T scanner. The device under study was a bent long wire 'antenna' made resonant at the scanner Larmor frequency with an adjustable series capacitor. The antenna, a 26 AWG (0.40 mm) Teflon-insulated silver-plated copper wire taped around the edge of the patient table, was terminated in a simulated RF ablation needle (a 15 cm long 16 AWG/1.30 mm diameter bare copper wire), the tip of which was embedded into the liver to simulate the percutaneous ablation of a solid hepatic tumor. The parameterized testbed coil model was used to replicate the RF antenna and needle geometry of *Hue et al., 2018*, using standard electrical properties of human liver tissue. The peak tissue temperature increase imaged in *Hue et al., 2018*, by the proton resonance frequency shift method (20°C) and a 22°C increase recorded by a fiber optic temperature sensor at the needle tip agreed well with our modeled prediction of a 23°C increase using the modeling testbed.

A very fine FEM mesh resolution is required to accurately resolve temperature rise close to a sharp lead tip with a diameter of 1.3 mm – *Figure 5*. This is achieved using local automated adaptive FEM mesh refinement, which is a unique property of the present testbed. *Figure 5* shows the corresponding testbed setup along with the lesion. A few relevant field movies are available as supplementary materials (*Online Dropbox, 2023*).

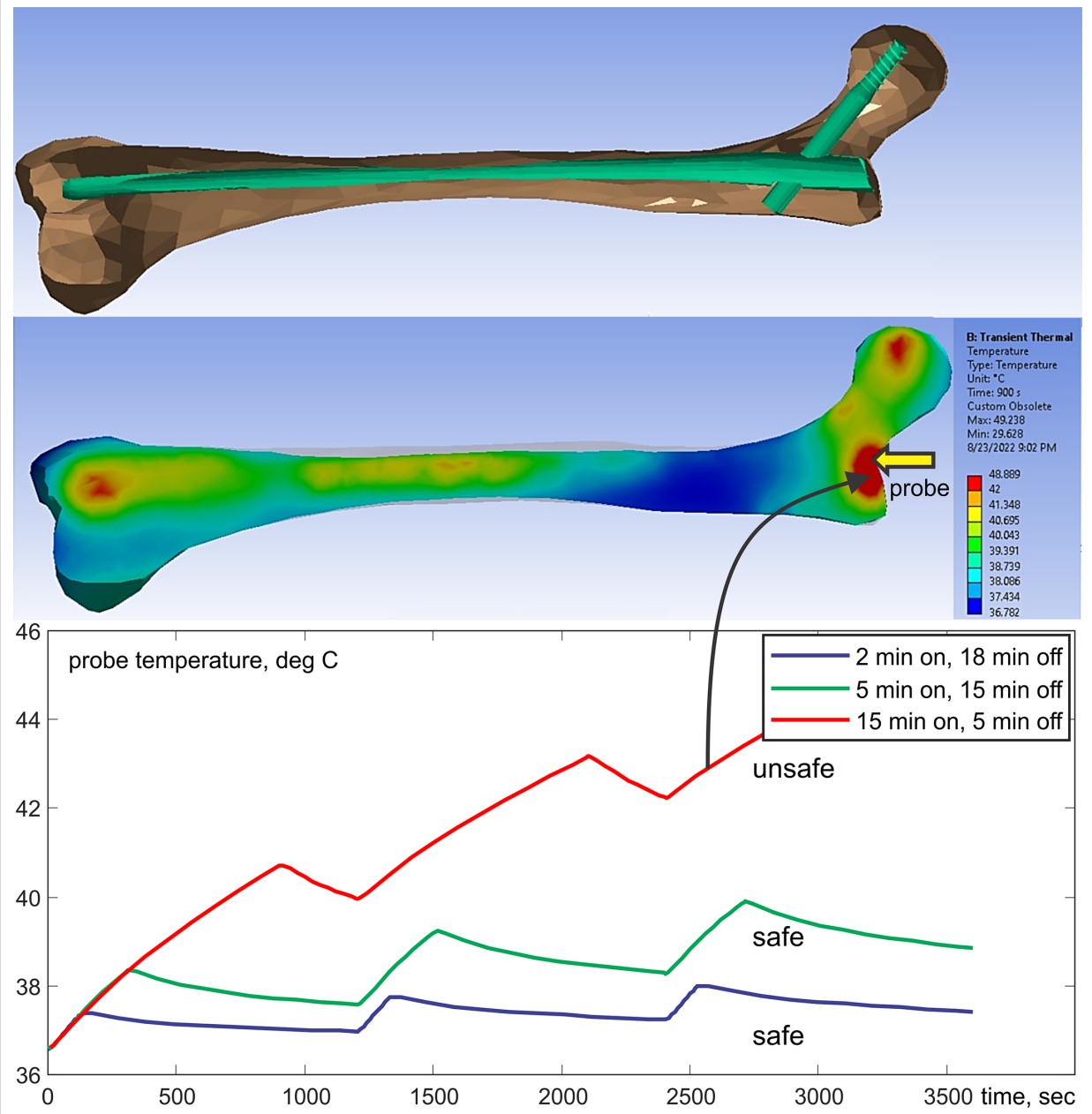

**Figure 4.** Top – long femoral nail subject to three repetitions of 15 min exposure followed by 5 min of rest for 1 hr in total. Center – temperature contour plot in a cut plane roughly bisecting the embedded femoral implant at the end of the last heating cycle. Bottom – temperature rise profile at the temperature probe. Only the bone is shown but the computations are performed for the entire model.

## Discussion and conclusion

The temperature rise in the surrounding tissues of a large orthopedic metallic implant subject to MR imaging is a significant point of concern and a potential barrier for the development of better implants. Numerical electromagnetic and thermal modeling offers a way to solve this complex problem with a sufficient degree of accuracy. We developed a complete testbed for realistic implant modeling, which includes a detailed FEM-compatible obese human female model, a parameterized tunable generic MR coil model, a method for implant embedding, and an accurate RF solver directly coupled with a transient thermal solver.

In the testbed, the MR exposure time is arbitrary and can be readily adjusted and rapidly tested (typically within ~5–7 min). This enables the user to meet regulatory requirements (**US Department**

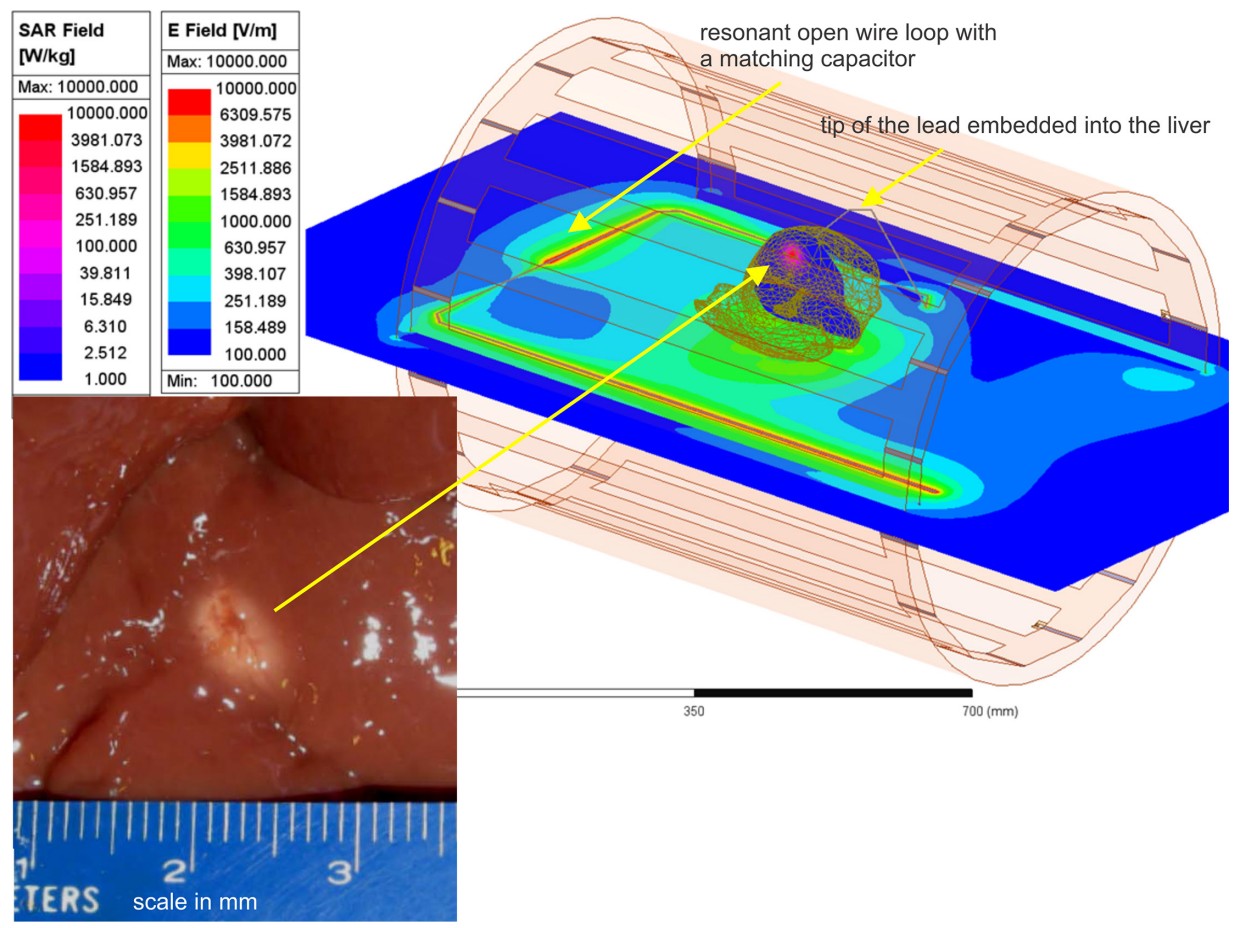

**Figure 5.** Liver experiment setup: E-field distribution in the plane of the resonant loop and specific absorption rate (SAR) distribution within liver in a plane passing through the tip of the lead. Bottom left – lesion in a section of ex vivo bovine liver created with heating at the tip of a 16 gauge (1.3 mm) bare copper wire needle in about 1.5 min (*Hue et al., 2018*).

of Health and Human Services et al., 2021) and to construct a proper MR exposure protocol. A cross-platform compatibility of the in silico model has been established previously (*Yanamadala et al., 2016*). Further validation of cross-platform electromagnetic and thermal coupling performance is currently underway with Dassault Systèmes CST Studio Suite software package. The testbed along with the open-source version of the human model VHP-Female College (*NEVA Electromagnetics, LLC, 2015*) is available online at https://doi.org/10.5061/dryad.2jm63xswt.

While the workflow presented herein establishes a validated approach to estimate RF heating due to the presence of a passive implant within a human subject undergoing an MR procedure, certain limitations and proper use stipulations of this methodology should be identified. These include:

1. The approach of embedding a given passive implant must be carefully considered and supervised by an orthopedic subject matter expert, preferably an orthopedic surgeon. While the procedures described above focus on insertion and registration of an implant to make it numerically suitable for simulation, relevant anatomic and physiological considerations must also be addressed to ensure a physically realistic and appropriate result. This will enable a proper simulated fit and no empty spaces or unintended tissue deformations.

2. Temperature changes presented are due only to RF energy deposition. The results do not take into account the impact of low-frequency induction heating of metallic implants naturally caused by switched gradient fields. Important work on this subject matter has recently been reported in *Winter et al., 2021*; *Wooldridge et al., 2021*; *Arduino et al., 2021*; *Arduino et al., 2022a*; *Bassen and Zaidi, 2022*; *Arduino et al., 2022b*; *Clementi et al., 2022*. Unless an orthopedic implant has a loop path, heating due to gradient fields is typically less than heating due to RF energy deposition. The present testbed would be applicable to the induction heating of

implants (and the expected temperature rise of nearby tissues), after switching from Ansys HFSS (the full wave electromagnetic FEM solver) to Ansys Maxwell (the eddy current FEM solver). Two examples of this kind have already been considered in *Bassen and Zaidi, 2022*; *Bryan David Stem, 2014*.

3. The procedures presented in this work have been based on the response of a single human model of advanced age and high morbidity.
4. Finally, validation was achieved using available published data (*Muranaka et al., 2006*; *Muranaka et al., 2007*; *Muranaka et al., 2011*) and relies upon the legitimacy and veracity of that data. Coil geometry, power settings, and other relevant parameters were taken explicitly from these sources and modeled to enable a faithful comparison.

## Acknowledgements

This work was supported by the National Institutes of Health (NIH) grants R01AR075077 and R01EB029818.

## Additional information

### Competing interests

Peter J Serano, Marc Horner: Employees of Ansys. Alexander Prokop: Employee of Dassault Systemes Deutschland GmbH. Jonathan Hanson: Employee of Neva Electromagnetics, LLC. Kyoko Fujimoto: Employee of GE HealthCare. James Brown: Employee of Micro Systems Enigineering, Inc. The other authors declare that no competing interests exist.

### Funding

| Funder | Grant reference number | Author |
| --- | --- | --- |
| National Institutes of Health | R01AR075077 | Jerome Ackerman |
| National Institutes of Health | R01EB029818 | Jerome Ackerman |

The funders had no role in study design, data collection and interpretation, or the decision to submit the work for publication.

### Author contributions

Gregory M Noetscher, Conceptualization, Formal analysis, Supervision, Validation, Investigation, Methodology, Writing – original draft, Project administration, Writing – review and editing; Peter J Serano, Software, Investigation, Methodology; Marc Horner, Conceptualization, Writing – original draft, Writing – review and editing; Alexander Prokop, Jonathan Hanson, Conceptualization, Writing – review and editing; Kyoko Fujimoto, Conceptualization, Supervision, Validation, Methodology, Writing – original draft, Writing – review and editing; James Brown, Ara Nazarian, Conceptualization, Supervision, Writing – original draft, Writing – review and editing; Jerome Ackerman, Conceptualization, Supervision, Investigation, Methodology, Writing – original draft, Writing – review and editing; Sergey N Makaroff, Conceptualization, Software, Formal analysis, Supervision, Validation, Investigation, Methodology, Writing – original draft, Project administration, Writing – review and editing

### Author ORCIDs

Gregory M Noetscher (ID) https://orcid.org/0000-0001-9786-7206
Marc Horner (ID) https://orcid.org/0000-0002-2483-5796
Jerome Ackerman (ID) http://orcid.org/0000-0001-5176-7496

Reviewer #1 (Public Review): https://doi.org/10.7554/eLife.90440.3.sa1
Reviewer #2 (Public Review): https://doi.org/10.7554/eLife.90440.3.sa2
Author Response https://doi.org/10.7554/eLife.90440.3.sa3

## Additional files

### Supplementary files
• MDAR checklist

### Data availability
Data supporting this study may be found here: https://doi.org/10.5061/dryad.2jm63xswt.

The following dataset was generated:

| Author(s) | Year | Dataset title | Dataset URL | Database and Identifier |
|---|---|---|---|---|
| Noetscher GM, Serano PJ, Horner M, Prokop A, Hanson J, Fujimoto K, Brown JE, Nazarian A, Ackerman J, Makaroff SN | 2023 | An In-Silico Testbed for Fast and Accurate MR Labeling of Orthopaedic Implants | https://doi.org/10.5061/dryad.2jm63xswt | Dryad, 10.5061/dryad.2jm63xswt |

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
