## [Editor Report · eLife assessment]

This manuscript will provide a **valuable** method to evaluate the safety of MR in patients with orthopaedic implants, which is required in clinics. A strength of the work is that the in-silicon testbed is **solid**, based on the widely available human project, and validated. In addition, the toolbox will be open for clinical practice.

---

## [Referee Report · Reviewer #1 (Public Review)]

Summary:

In this work authors are trying to satisfy a real need in MR safety, when concerns can rise about the thermal increase due to metallic materials in patients carrying orthopedic implants. The "MR conditional" labeling of the implant obtained by ASTM in-vitro tests may help to plan the MR scan, but it is normally limited to a single specific MR sequence and a B0 value, and it is not always available. The adoption of an in-silico simulation testbed overcomes this limitation, providing a fast and reliable prediction of temperature increase from RF, in real-life scan conditions on human-like digital models. The FDA is pushing this approach.

Strengths:

The presented in-silico testbed looks valuable and validated. It is based on the widely available Visible Human Project (VHP) datasets, and the testbed is available on-line. The approval of the testbed by the FDA as a medical device development tool (MDDT) is a good premise for the large-scale adoption of this kind of solution.

Weaknesses:

A couple of limitations of the study are now clearly highlighted to the readers in this revised version of the paper. The following aspects:

- the lack of the equivalent modeling for the gradients-related heating;

- the way the implant is embedded in the VHP model that should take in consideration how to manage the removed and stretched tissues;

are now correctly taken in consideration in the discussion, providing additional literature.

---

## [Referee Report · Reviewer #2 (Public Review)]

Summary:

In this article, the authors provide a method of evaluating safety of orthopedic implants in relation to Radiofrequency induced heating issues. The authors provide an open source computational heterogeneous human model and explain computational techniques in a finite element method solver to predict the RF induced temperature increase due to an orthopedic implant while being exposed to MRI RF fields at 1.5 T.

Strengths:

The open access computational human model along with their semiautomatic algorithm to position the implant can help realistically model the implant RF exposure in patient avoiding over- or under-estimation of RF heating measured using rectangular box phantoms such as ASTM phantom. Additionally, using numerical simulation to predict radiofrequency induced heating will be much easier compared to the experimental measurements in MRI scanner, especially when the scanner availability is limited.

Weaknesses:

The proposed method only used radiofrequency (RF) field exposure to evaluate the heating around the implant. However, in the case of bulky implants the rapidly changing gradient field can also produce significant heating due to large eddy currents. So the gradient induced heating still remains an issue to be evaluated to decide on the safety of the patient. Moreover, the method is limited to a single human model and might not be representative of patients with different age, sex and body weights.

---

## [Author Response]

The following is the authors’ response to the original reviews.

The Authors wish to thank the Reviewers for their detailed and insightful comments. By properly addressing these critiques, we sincerely believe our finished product will be substantially improved and provide greater insight to the academic community.

Both Reviewers noted the importance of identifying the limitations of our study with particular emphasis on embedded implant heating due to switching gradient coils. Understanding the limitations of any model and/or simulation process is critical when adopting its use, especially when estimating the safety of embedded devices. For this reason, we have included the following text and corresponding references in our Discussion section:

While the workflow presented herein establishes a validated approach to estimate RF heating due to the presence of a passive implant within a human subject undergoing an MR procedure, certain limitations and proper use stipulations of this methodology should be identified. These include:

1. The approach of embedding a given passive implant must be carefully considered and supervised by an orthopaedic subject matter expert, preferably an orthopaedic surgeon. While the procedures described above focus on insertion and registration of an implant to make it numerically suitable for simulation, relevant anatomic and physiological considerations must also be addressed to ensure a physically realistic and appropriate result. This will enable a proper simulated fit and no empty spaces or unintended tissue deformations.

2. Temperature changes presented are due only to RF energy deposition. The results do not take into account the impact of low-frequency induction heating of metallic implants naturally caused by the switching gradient coils. Important work on this subject matter has recently been reported in [21],[22],[23],[24],[25],[26],[27]. Unless an orthopaedic implant has a loop path, heating due to gradient fields is typically less than heating due to RF energy deposition. The present testbed would be applicable to the induction heating of implants (and the expected temperature rise of nearby tissues), after switching from Ansys HFSS (the full wave electromagnetic FEM solver) to Ansys Maxwell (the eddy current FEM solver). Two examples of this kind have already been considered in [25],[45].

3. The procedures presented in this work have been based on the response of a single human model of advanced age and high morbidity.

4. Finally, validation was achieved using available published data [42]-[44] and relies upon the legitimacy and veracity of that data. Coil geometry, power settings, and other relevant parameters were taken explicitly from these sources and modeled to enable a faithful comparison.